# Transcriptomic meta-signatures identified in *Anopheles gambiae* populations reveal previously undetected insecticide resistance mechanisms

V.A. Ingham [1], S. Wagstaff[2] & H. Ranson[1]

Increasing insecticide resistance in malaria-transmitting vectors represents a public health threat, but underlying mechanisms are poorly understood. Here, a data integration approach is used to analyse transcriptomic data from comparisons of insecticide resistant and susceptible *Anopheles* populations from disparate geographical regions across the African continent. An unbiased, integrated analysis of this data confirms previously described resistance candidates but also identifies multiple novel genes involving alternative resistance mechanisms, including sequestration, and transcription factors regulating multiple down-stream effector genes, which are validated by gene silencing. The integrated datasets can be interrogated with a bespoke Shiny R script, deployed as an interactive web-based application, that maps the expression of resistance candidates and identifies co-regulated transcripts that may give clues to the function of novel resistance-associated genes.

[1] Vector Biology, Liverpool School of Tropical Medicine, Pembroke Place, Liverpool L35QA, UK. [2] Bioinformatics Unit, Liverpool School of Tropical Medicine, Pembroke Place, Liverpool L35QA, UK. Correspondence and requests for materials should be addressed to V.A.I. (email: victoria.ingham@lstmed.ac.uk)

Insecticide resistance is a major threat to global public health, reducing the efficacy of vector control efforts, which currently rely heavily on four public health insecticides for use in insecticide impregnated materials and in indoor residual or aerial spraying applications[1,2]. Resistance to all these insecticide classes is widespread in major disease vectors and there are increasing reports of vector control breakdown as a result of this resistance[1,3–6]. Understanding the causes of this resistance is critical for developing resistance management strategies and to inform the development of new public health insecticides[1,3,4]. A wealth of data exists on the gene expression patterns of insecticide resistant *Anopheles* malaria vector populations and corresponding susceptible populations from across Sub-Saharan Africa[7–18]. Analysis of such datasets yield lists of hundreds to thousands of transcripts showing different expression patterns in resistant mosquitoes. In many of these studies, only a priori candidate genes such as those encoding enzymes from families with known insecticide detoxification functions, have been chosen for further analysis and validation[19–22], reducing complex and informative datasets to patterns in known gene families. Identifying patterns of expression in transcripts across multiple datasets may offer a deeper insight into the mechanisms of insecticide resistance by finding intersecting processes across independent but related datasets. However, recognising these patterns for insecticide resistance has been confounded by the complexities of the data available, such as variable susceptible reference strains, broad geographical distances between populations analysed, and differing experimental designs.

Here, 31 datasets comparing resistant and susceptible populations from three of the major malaria vector species in Africa, *Anopheles gambiae*, *An. coluzzii* and *An. arabiensis* were retrieved from published literature[7–18] (Fig. 1) and re-analysed using the limma[23,24] package in R. The datasets represent mosquito populations from a disparate geographical range, covering much of Sub-Saharan Africa and span five years of collections. We searched these data for meta-signatures associated with insecticide resistance, which resulted in several unreported potential mechanisms, both at a regulatory and direct mechanistic level. Firstly, we identified a number of transcripts consistently up-regulated in pyrethroid resistant *An. gambiae* s.l. populations across Africa. Most of these transcripts belong to gene families not previously associated with insecticide resistance but silencing expression of a subset of these genes in resistant populations significantly supressed the resistant phenotype. Secondly, we identified two additional transcription factors that regulate expression of genes associated with insecticide resistance. Finally, we developed a bespoke web-based application (app) in ShinyR[25], IR-TEx (Insecticide Resistance Transcript Explorer), that enables all users to explore transcripts of interest, map their associated expression across Africa, and identify putative functions and pathways of their transcripts using pairwise correlation matrices.

## Results

**Available datasets.** ArrayExpress[26] and VectorBase[27] were searched for available microarray datasets comparing resistant and susceptible mosquitoes from the three dominant vector species in the *Anopheles gambiae* complex: *An. gambiae*, *An. coluzzii* and *An. arabiensis*. These members of the *An. gambiae* species complex have overlapping distributions; however, although hybrids are viable, they are primarily found at low frequency in wild populations, indicating limited introgression, with the exception of the pyrethroid target site mutation, *kdr*[28]. From this search, datasets were retrieved from 22 geographically distinct mosquito populations (Fig. 1);[7–18] the total number of datasets available for this study was 31 as mosquito collections from some

sites were used in multiple experiments either (i) involving comparisons to different susceptible populations or (ii) as part of a temporal study. These represent disparate datasets collected from across regions of the continent with the highest malaria endemicity and span 5 years of collections[7–18]. Meta-data, including resistance status, insecticide exposure and *kdr* frequency for the 31 datasets used in this study are provided in Supplementary Table 1. Initially, the metadata itself was used to cluster the microarray datasets to identify patterns of resistance in population subsets, but as this only explained <0.5% of the variation, we instead took hypothesis-based approach to identify patterns across all data sets.

**Candidate gene families in pyrethroid resistant mosquitoes.** As resistance to pyrethroid insecticides, the only insecticide class currently used to treat bednets is of most immediate threat to malaria control, our analysis focused on populations resistant to this insecticide class. Of the 31 datasets, 12 compared gene expression in mosquitoes that had been exposed to pyrethroids (with RNA extracted from survivors 48 h post-exposure) with a susceptible unexposed population (dashed circles, Fig. 1). A total of 101 transcripts from 86 genes showed the same direction of differential expression (i.e. higher in resistant population, herein referred to as up-regulated, or higher in susceptible population, referred to as down-regulated) across all 12 datasets; 56 genes were up-regulated and 30 genes were down-regulated (Supplementary Table 2). These data represent pyrethroid resistant populations from seven countries and include five populations of *An. coluzzii*, one *An. gambiae* and three *An. arabiensis* (two *An. arabiensis* and one *An. coluzzii* population were compared to two separate susceptible strains hence 12 datasets were included in total (see Supplementary Table 1 for further details on the mosquito populations)).

**Candidate gene families in pyrethroid resistant *An. coluzzii*.** Data were further subdivided into species as the low levels of introgression between the *An. gambiae* species complex increases the likelihood of species specific resistance mechanisms. Analysis of the 11 highly or moderately pyrethroid resistant (unexposed or exposed) *An. coluzzii* populations identified a total of 43 transcripts from 41 genes that were significantly differentially expressed in the same direction (compared to lab susceptible populations) across each experimental set (Supplementary Table 3). Despite many of the same populations being used to produce the gene lists in Supplementary Table 2 and Supplementary Table 3, with the notable exception of the α-crystallin family, there is little overlap in transcript identity between these two lists (just two transcripts are commonly up-regulated across both tables).

**Detoxification candidates.** Enhanced detoxification of insecticides is thought to be one of the major resistance mechanisms and several glutathione transferases (GSTs), cytochrome P450s (CYPs) and carboxylesterases (COEs) have previously been shown to be elevated in resistant population and encode enzymes that detoxify insecticides[7–18]. Nine of the 56 genes commonly up-regulated in resistant populations across the species complex belong to these three gene families. This list includes two known pyrethroid metabolisers, *CYP6Z3* (AGAP008217)[20] and *GSTD1* (AGAP004164)[29], but also seven additional detoxification genes, (*GSTD7* (AGAP004163), *GSTD3* (AGAP004382), *GSTE5* (AGAP009192), *GSTMS3* (AGAP009946), *COEAE8O* (AGAP006700), *CYP4C28* (AGAP010414) and *CYP12F2* (AGAP008020) (note that the *CYP12F2* probe overlaps with the unnamed p450 AGAP012800 (Supplementary Table 4)) which,

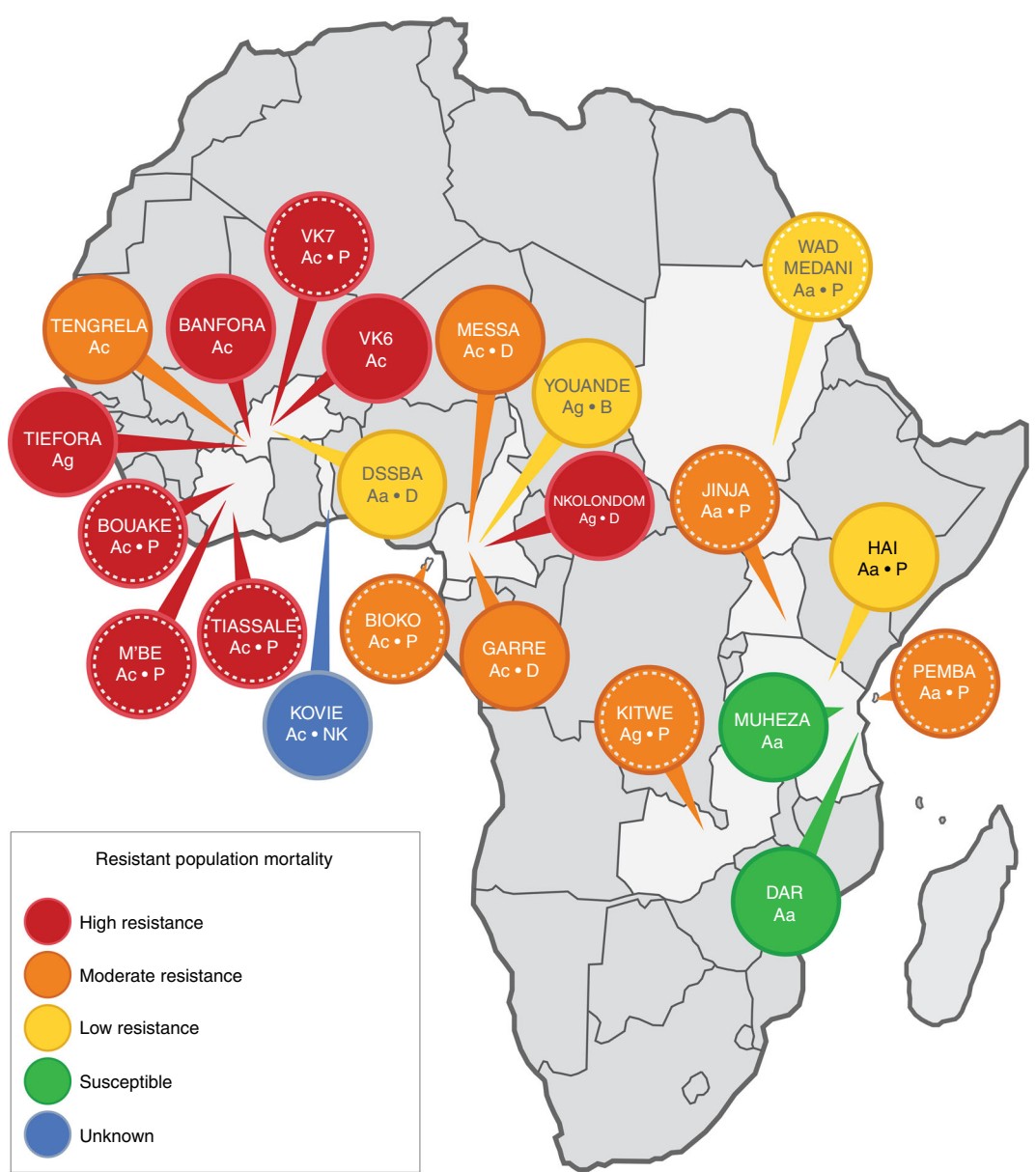

**Fig. 1** Distribution of available microarray datasets. Microarray datasets available from sub-Saharan Africa comparing insecticide resistant and susceptible: *An. gambiae* (Ag), *An. coluzzii* (Ac) or *An. arabiensis* (Aa). Resistance levels are characterised by the populations maximal recorded mortality in WHO discriminating dose assays: high resistance 0–33% mortality, moderate resistance 33–66%, low resistance 66–90% mortality, susceptible populations are those that consistently exhibited 100% mortality after exposure (See Supplementary Table 1). The insecticides that the populations have been exposed to prior to RNA extraction are represented by D DDT, P Pyrethroid, B Bendiocarb, NK Not known; unexposed mosquitoes have no corresponding letter. Figure created expressly for this manuscript by Manuela Bernardi

as far as we are aware, have not been evaluated for insecticide metabolising activity. The cytochrome P450 cofactor, cytochrome b5 (AGAP007121), is also amongst this list of genes commonly upregulated in pyrethroid resistant populations. *CYP4D22* (AGAP002419) is the only candidate detoxification gene consistently up-regulated across all 11 *An. coluzzii* populations.

Several of the better characterised insecticide detoxification genes, such as *CYP6M2* (AGAP008212)[19,30] and *CYP6P3* (AGAP002865)[22] are not found in this list of 56 common genes up-regulated in all pyrethroid resistant populations of *An. gambiae* s.l. although these are represented in multiple individual populations. A heatmap showing the expression of the 11 cytochrome P450s (10 % of the total family in this species) most commonly associated with resistance is provided in

Supplementary Figure 1. Cognisant that genome sequencing of several hundred *An. gambiae* individuals from across Africa has pointed to multiple origins of insecticide resistance[31] our approach is not intended to identify candidates that emerge from single populations. Instead, our data integration approach of pooling populations from diverse geographical origins is aimed at identifying putative candidates that have previously been overlooked by studies on single populations due to a priori candidate focus and to highlight common biological and cellular mechanisms operating at a higher level.

**Novel gene families associated with pyrethroid resistance.** Seven genes from Supplementary Tables 2 and 3 were selected for qPCR validation in three pyrethroid resistant colonies of *An.*

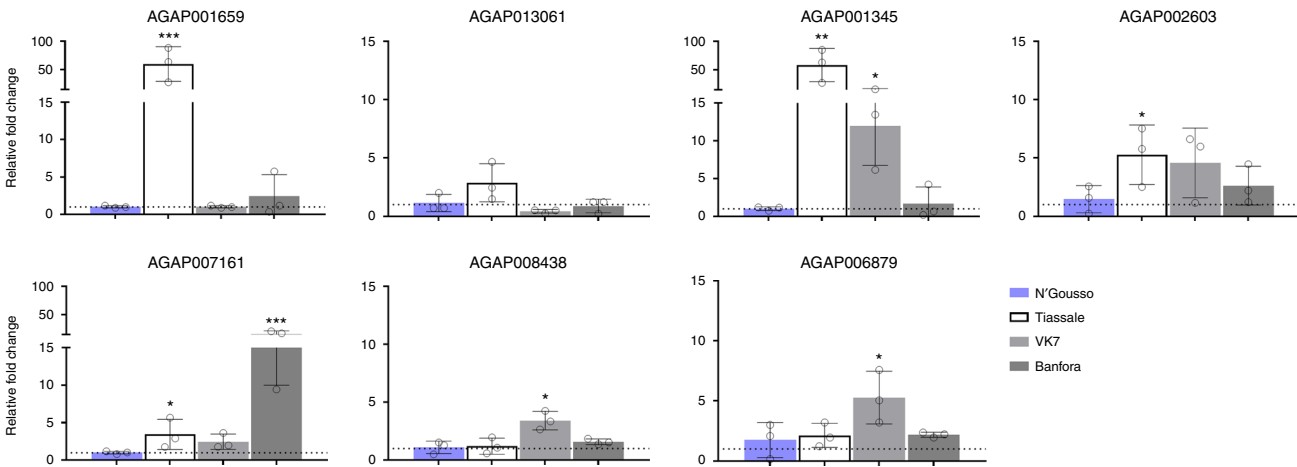

**Fig. 2** qPCR analysis of top insecticide resistance candidates from multiple resistant populations. qPCR results of three 3–5 day old, unexposed, pyrethroid resistant *Anopheles* populations compared to the lab susceptible N'Gousso, three biological replicates and three technical replicates were used for each gene. Relative fold change ($y$), and resistant populations ($x$). Standard deviation bars are shown, with significance $*p \leq 0.05$, $**p \leq 0.01$, $***p \leq 0.001$ as calculated by ANOVA with Dunnett's post hoc test

*gambiae* s.l. (Tiassalé, VK7 and Banfora) maintained at LSTM. Six were significantly over expressed in at least one of the resistant colonies compared to the laboratory susceptible N'Gousso strain (Fig. 2). (It should be noted that mosquitoes used in the qPCR had not been exposed to insecticides; had we-pre-exposed to pyrethroids, as was the case in the majority of the microarray experiments, an even stronger correlation might have been expected.) The hexamerin and α-crystallin families, plus the ATPase subunit (as it had the highest mean fold change of all candidate transcripts) were selected for further validation.

**α-crystallins**. Amongst the top five genes up-regulated across pyrethroid resistant populations (Supplementary Table 2) and confirmed by qPCR on lab colonies (Fig. 2) is a member of the α-crystallin family, AGAP007161. α-crystallins are small, ATP-independent chaperone proteins, that are induced by a variety of stresses including heat stress, hyperoxia and oxidative stress;[32] they directly bind to stress induced mis-folded proteins, preventing toxic aggregation[33]. AGAP007161 is one of eight putative α-crystallins in *An. gambiae*, five of which are clustered on chromosome 2 L, division 27B in the genome; transcripts from this cluster of paralogous genes are up-regulated across multiple microarray datasets, indicating possible transcriptional co-regulation (Supplementary Figure 2). qPCR characterisation of their expression confirmed that four of the five 2 L α-crystallins are overexpressed in at least one of the three pyrethroid resistant laboratory colonies (Supplementary Figure 3).

As α-crystallins are frequently up-regulated in response to stress, the impact of insecticide exposure on the expression of the four members of this gene family constitutively overexpressed in resistant populations was investigated. AGAP007159 was strongly induced 24 and 48 h post deltamethrin exposure (relative fold changes, compared to unexposed Tiassalé control = 22.1x and 60.5x respectively) whereas other members of this gene cluster showed reduced expression after exposure to insecticides (Supplementary Figure 4). AGAP007159 was not one of members of the α-crystallin family over expressed in the microarray analysis (Supplementary Tables 2 or 3) but results for this transcript may be confounded by the cross hybridisation between the probe for AGAP007159 and other member of this gene family (Supplementary Table 4), which can then be discerned by unique primer sets in qPCR.

Four of the α-crystallins were silenced in one or more pyrethroid resistant strains and the impact on the resistance phenotype investigated; in each case no difference in mortality after gene knockdown was seen after exposure to control papers, indicating no dramatic short-term fitness cost of gene silencing. Attenuation of expression of AGAP007159 by RNAi resulted in a large significant increase in mortality following deltamethrin exposure in two pyrethroid resistant lab colonies (43.5% mortality in dsGFP controls compared to 82.3% in Tiassalé, $p_{(ANOVA,\ Tukey\ post\ hoc)} = 0.0003$, 19.4% in dsGFP compared to 62.4% in VK7, $p_{(ANOVA,\ Tukey\ post\ hoc)} = 0.0002$). Silencing of the non-induced α-crystallins had no effect on pyrethroid induced mortality; this could be due to the ubiquitously over-expressed a-crystallins having functional redundancy or a non-crucial component in a resistance pathway (Fig. 3).

**Hexamerins**. The list of transcripts up-regulated across pyrethroid resistant populations contained multiple members of the hexamerin family (Supplementary Table 2). These proteins are the most abundant proteins in the haemolymph, where they act as storage and transport proteins[34]. There are eight putative hexamerins in *An. gambiae* and six of these are up-regulated across multiple pyrethroid resistant microarray datasets (Supplementary Figure 5) (AGAP005766 and AGAP005767 have identical probes, so these transcripts will experience cross-hybridisation on the arrays) with qPCR confirming that five of these are up-regulated in at least one of the three pyrethroid resistant lab colonies (Supplementary Figure 6). RNAi was again used to determine whether up-regulation of these hexamerin transcripts was associated with pyrethroid resistance. dsRNA mediated attenuation of AGAP001659 resulted in a small but significant increase (43.5–60.1% $p_{(ANOVA,\ Tukey\ post\ hoc)} = 0.0415$) in mortality after deltamethrin exposure when compared to GFP controls in the Tiassalé strain although this phenotype was not observed in VK7 (Fig. 3). Suppression of expression of AGAP001657 or AGAP001345 had no significant impact on the pyrethroid resistant phenotype.

**ATPase subunit e**. The final gene from the meta-analysis selected for RNAi validation was AGAP006879, which encodes subunit e of the F-type ATP synthase. This transcript was the most highly

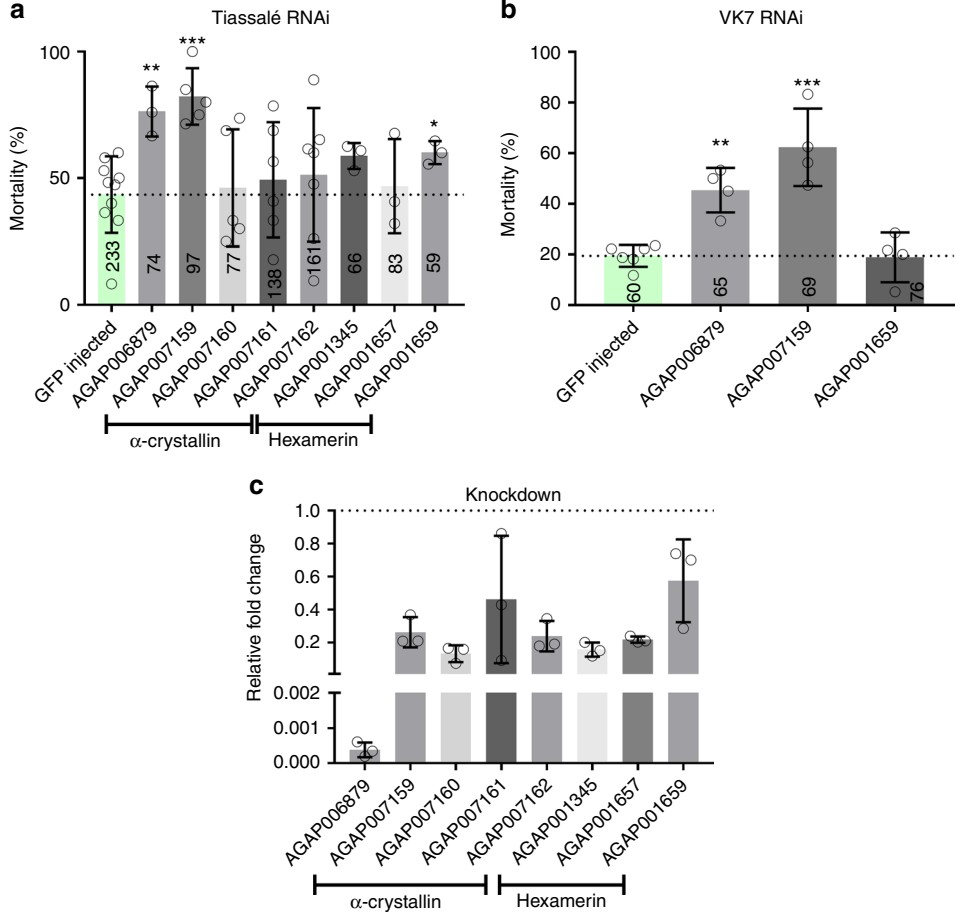

**Fig. 3** RNAi phenotyping for potential candidates. RNAi was performed using two pyrethroid resistant colonies, Tiassalé and VK7. 3–5 day old females mosquitoes were injected with dsRNA from each of the respective transcripts. 72-h post injection, the mosquitoes were exposed to 0.05% WHO deltamethrin papers for 1 h and mortality scored 24 h later. **a** Mortality associated with dsRNA induced knockdown in Tiassalé mosquitoes. **b** dsRNAs showing significantly increased mortality in Tiassalé mosquitoes were then injected into a second resistant colony, VK7. **c** Knockdown levels for each dsRNA construct relative to GFP-injected controls (Tiassalé population only). Error bars represent standard deviation, three biological replicates and three technical replicates were used for each gene with each circle representing the mean value for each biological replicate in the qPCR data and significance is indicated by *$p \leq 0.05$, **$p \leq 0.01$, ***$p \leq 0.001$, as computed by ANOVA with post hoc Tukey correction. Numbers on the bars represent number of mosquitoes tested under each condition

overexpressed across all An. coluzzii arrays ($FC_\mu = 37.26$, Supplementary Table 3) and its over expression in lab colonies was confirmed by qPCR (Fig. 2). Suppressing expression of the ATPase resulted in a significant increase in mortality in both Tiassalé and VK7 mosquitoes post deltamethrin exposure (43.46–76.34%, $p_{(ANOVA, Tukey\ post\ hoc)} = 0.0099$, 19.4–45.4%, $p_{(ANOVA, Tukey\ post\ hoc)} = 0.00146$ respectively) (Fig. 3). Although previously thought to only have a role in the mitochondria, ATPases are present in the plasma membrane of the insect midgut and salivary glands and have been shown to have a role in lipid transport[35,36].

**Transcriptional regulation of insecticide resistance.** Silencing of the transcription factor *Maf-S* has recently been shown to modulate expression of key insecticide detoxification genes[37], providing evidence for a major role in the regulation of metabolic resistance. Screening the previously published transcriptomic data set on ds*Maf-S* compared to ds*GFP* control revealed that 30% of the pyrethroid resistance candidates identified in the current study, including AGAP002603 (EF-like Factor), AGAP001659 (Hexamerin), AGAP008217 (*CYP6Z3*) and two α-crystallins (AGAP007161 and AGAP007160) show decreased expression when the *Maf-S* transcription factor is silenced, providing further

evidence for the importance of *Maf-S* in controlling insecticide resistance (Supplementary Table 5).

In order to identify other potential regulatory pathways, putative homologues of all 560 transcription factors described in *Drosophila melanogaster*[38] were identified using FlyMine[39] and searched against the *An. gambiae* s.l. datasets. Twenty-five transcription factors differentially expressed in at least 50% of the data sets from insecticide resistant compared to susceptible populations were identified, including *Maf-S* (Supplementary Figure 7). Literature searches show that five of these transcription factors are linked to stress responses: *Met* has been indirectly linked to stress response to insecticide exposure in *Drosophila*;[40] *AP-1* (*jra*)[41] and *TFAM*[42] are linked to oxidative stress; *sug* has been linked to salt[43], starvation and sugar stress;[44] and *REL2* has been directly implicated in response to permethrin exposure[45]. Two of the 25 transcripts, *Dm* (previously unlinked to stress) and *Met*, were individually suppressed by RNAi in the resistant Tiassalé strain and both led to a significant increase in mortality ($p_{(ANOVA, Tukey\ post\ hoc)} = 0.022$, $p_{(ANOVA, Tukey\ post\ hoc)} = 0.00098$) after exposure to deltamethrin (Fig. 4).

In order to determine whether this data integration approach was a valid method for identifying transcription factors controlling expression of insecticide resistance candidates, a microarray

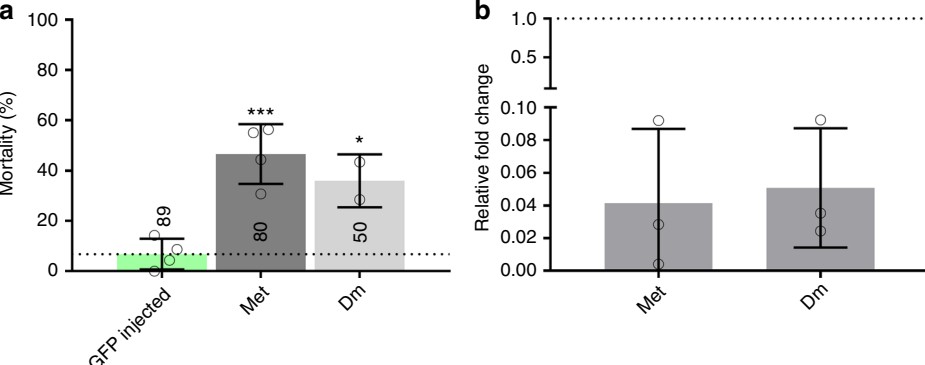

**Fig. 4** RNAi phenotyping for potential transcription factor candidates. RNAi was performed using the pyrethroid resistant Tiassalé colony. Three to five-day-old females mosquitoes were injected with dsRNA from each of the respective transcripts. Seventy two-hours post injection, the mosquitoes were exposed to 0.05% WHO deltamethrin papers for 1 h and mortality scored 24 h later. **a** Deltamethrin induced mortality following dsRNA induced knockdown of the transcription factors Met and Dm. **b** Knockdown levels for each dsRNA construct relative to GFP-injected controls. Error bars represent standard deviation, three biological replicates and three associated technical replicates were used for each gene in the qPCR data and significance is indicated by $*p \leq 0.05$, $**p \leq 0.01$, $***p \leq 0.001$, as computed by ANOVA with post hoc Tukey correction. Numbers on the bars represent number of mosquitoes tested under each condition

experiment was performed to compare gene expression in ds*Met* and ds*GFP* (control) female mosquitoes. A total of 1365 transcripts were differentially expressed between the two test sets (Supplementary Data 1); this dataset is described in more detail in ArrayExpress (E-MTAB-4043). Up-regulated transcripts from the *Met* knockdown experiment were significantly enriched for Gene Ontology (GO) terms related to key housekeeping and various binding activities whereas down-regulated transcripts were enriched for oxidation reduction processes, structural assembly, haem binding and transferase activity as seen in *Aedes aegypti*[46]. Only six detoxification candidates were up-regulated in the ds*Met* arrays (6 out of 688, no enrichment $p_{(\text{hypergeometric test})} = 0.95$). Conversely, there was significant enrichment (27 out of 677) for downregulated detoxification family members ($p_{(\text{hypergeometric test})} = 9.73\text{e-7}$). Fifteen of these down-regulated transcripts following *Met* transcription factor silencing are cytochrome p450s, including the insecticide metaboliser *CYP6M2* (AGAP008212)[19,30], consistent with *Met* having an activating role in transcription of these detoxification genes (Supplementary Table 6).

Down-regulated transcripts were enriched in vitellogenin and lipoprotein domains, concurrent with a role in ovarian development[46–48]. Similarly using the DroPhEA database[49] to determine putative phenotypic roles of enriched transcripts, 249 homologues had an effect on fertility ($p = 0.043$, as calculated by DroPhEA[49]), again consistent with a role in reproduction[46–48].

**App development**. The Insecticide Resistance Transcript Explorer (IR-TEx) was developed in ShinyR[25,50] to facilitate exploration, via an interactive browser applet of all insecticide resistance microarray datasets [https://www.lstmed.ac.uk/projects/ir-tex]. The aim of the app is to allow end-users to apply their own filtering criteria to identify whether genes of interest are differentially expressed in insecticide resistant populations and view the geographical distribution of populations differentially expressing these genes. Users can also input their own data sets to look for similarities or differences with other populations (Supplementary Note 1). IR-TEx displays the fold change of the transcript in all selected arrays in graphical (log₂) (Fig. 5a) and tabular form (raw); the user can save the graphical output and download the tabular output tab separated value format, which also contains adjusted $p$-values (Q values) for each dataset and the total number of datasets in which the transcript is

significantly differentially expressed. A map is also displayed for each transcript (Fig. 5b), allowing visualisation of the geographical distribution of the significance of the entered transcript and the associated fold change. IR-TEx also enables correlation networks to be visualised (Fig. 5c); by identifying co-regulated transcripts, putative pathways can be constructed and hypotheses on transcript function developed. Analogous to the transcript expression outputs, IR-TEx graphically displays the log₂ fold change of all transcripts fitting with the user-defined cut-off (defined by Pearson's $r$ value); a tabular version can be downloaded to explore the transcripts further (Supplementary Note 1).

**Using IR-TEx to predict gene function**. In addition to displaying fold change information graphically and geographically across populations, a powerful application of IR-TEx is to assign putative functions to transcripts through correlation networks; these transcripts could be (i) co-regulated or (ii) active in a single pathway. To illustrate this, the correlation networks of two genes that have previously been implicated in insecticide resistance, but are not involved in insecticide detoxification are described.

*CYP4G16* is a cytochrome p450 responsible for the decarbonylase step in cuticular hydrocarbon (CHC) synthesis[51]. Using a strict correlation cut-off of Pearson's pairwise $|r| \geq 0.85$, 44 transcripts (including all 4 splice variants of *CYP4G16* and the paralog *CYP4G17*) were found correlated with *CYP4G16* across the 31 datasets of insecticide resistant populations (Fig. 5c). GO term enrichment analysis of this list shows that the transcripts are lipid-related (GO terms: fatty acid biosynthetic process ($p = 5\text{e-}3$); 3-oxo-X-coA synthase activity (2.8e-3); hydrolase activity, acting on ester bonds ($p = 4.6\text{e-}3$); as calculated by DAVID enrichment[52]). Transcripts in this list include propionyl CoA synthase (AGAP001473), 1 fatty acid synthase, 2 fatty acid elongases, 1 fatty acid reductase and 1 fatty acid desaturase (Supplementary Table 7) all of which belong to enzyme families known to catalyse key steps in the CHC pathway (Fig. 6). Further evidence in support of the role of these genes in the CHC biosynthesis pathway is provided by data on tissue expression which shows 80% (35/44) are significantly enriched in the abdomen carcass of Tiassalé[53] (Supplementary Table 7), the site of the oenocyte cells which are responsible for CHC production[51]. This correlation network thus provides a basis for predicting

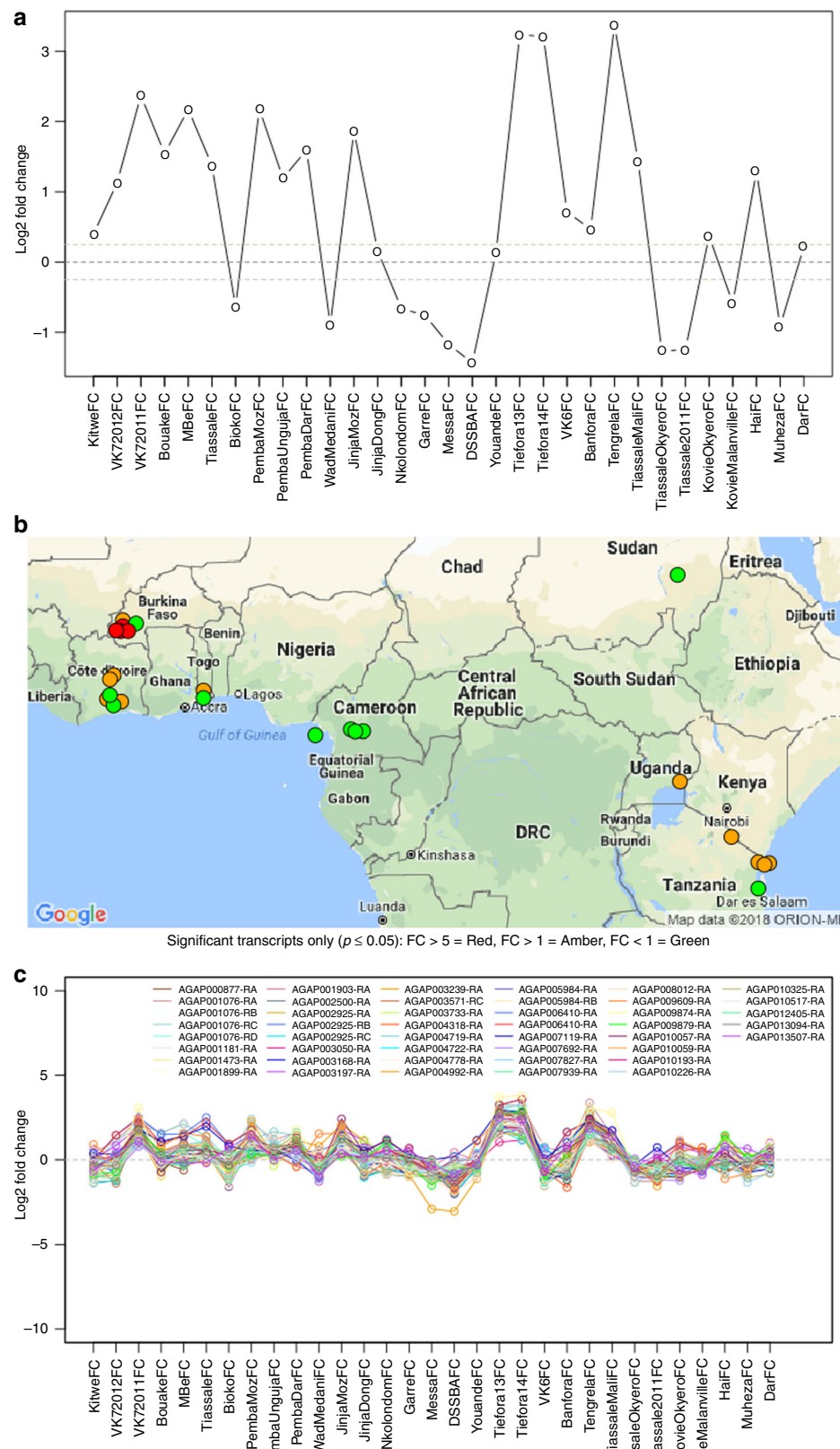

Significant transcripts only ($p \leq 0.05$): FC > 5 = Red, FC > 1 = Amber, FC < 1 = Green

specific genes involved in this biosynthetic pathway for functional validation.

Recently, the overexpression of multiple ABC transporters in pyrethroid resistant populations of *An. gambiae* was reported, along with the observation that several members of one of the larger subfamilies of this transporter family, the ABCGs, were enriched in the legs[54]. The correlation network ($|r| \geq 0.8$) provides clues as to a putative function for at least one member of this subfamily. *ABCG16* (AGAP009467) is correlated with just 5 transcripts; AGAP001763 a fatty acid transporter, AGAP003600 a

**Fig. 5** Graphical Outputs from IR-TEx App. These panels represent three graphical outputs from the IR-TEx App, each of which is available in downloadable tabular formats. These outputs result from a user inputted VectorBase ID, selected filtering criteria based on meta-data and finally, user inputted correlation value. Each panel here is AGAP001076-RA (*CYP4G16*). **a** Log$_2$ fold change of *CYP4G16* (*y*) across each microarray experiment meeting pre-selected user criteria (*x*), here all data is selected. **b** Each point represents a dataset with significant differential expression of *CYP4G16* and the associated approximate collection site. Green points show significant down regulation of *CYP4G16*, orange points show significant fold changes of 1-5 and red a fold change of >5 compared to susceptible controls. Map data source: Google Maps, 2018. **c** Log$_2$ fold change (*y*) across each microarray experiment meeting pre-selected user criteria (*x*) for each transcript showing a correlation of $|r| \geq 0.85$ for *CYP4G16*

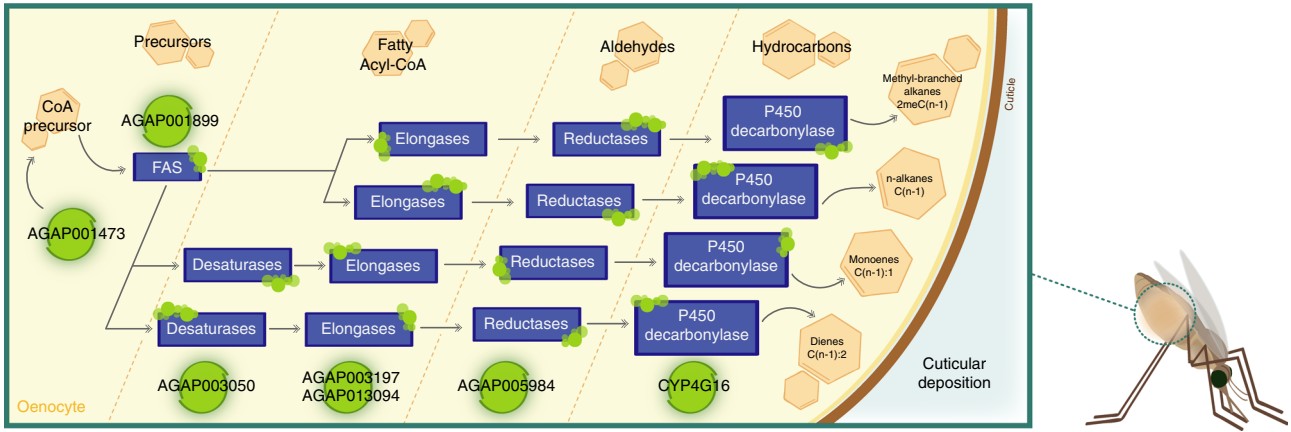

**Fig. 6** Schematic of cuticular hydrocarbon synthesis in *An. gambiae*. Schematic showing the steps in cuticular hydrocarbon synthesis, adapted from Blomquist 1987[67] and Balabanidou et al.[51], together with a list of putative members of each gene family in *An. gambiae*. All gene families are shown in blue and represent: FAS fatty acid synthases, elongases fatty acid elongases, reductases fatty acid reductases, desaturases fatty acid desaturases. General chemical classes at each step are shown in light red hexagons. Highlighted in green circles are the transcripts present in the correlation network of *CYP4G16*. The correlation network is significantly enriched for this pathway $p_{(\text{hypergeometric test})} = 1.44e^{-12}$. Figure created expressly for this manuscript by Manuela Bernardi

fatty acid elongase, AGAP005634 a chitinase, AGAP002739 an ortholog of *Drosophila* uninflatable protein which contains a LDL-receptor domain and AGAP010295 a Ca$^{2+}$ binding protein. The ABCG family has been implicated in lipid transport from the epidermis to the cuticle in other insect species[55,56] and the correlation network for *ABCG16* is certainly supportive of an orthologous function in *An. gambiae*.

## Discussion

Despite the large amount of data available on the transcriptomes of insecticide resistant mosquitoes from across sub-Saharan Africa, and the importance of this information for malaria control programmes, no concerted effort to analyse these data in an overarching manner has previously been reported. Using meta-signatures across populations with pyrethroid resistance profiles, several new and potent insecticide resistance gene candidates have been identified, acting across multiple resistant populations sourced from large geographical distances.

The candidates identified by this approach do not belong to gene families previously associated with pyrethroid resistance. Interestingly, two of the new gene families implicated, the hexamerins and α-crystallins, are typically associated with binding and storage roles[33,34] and may hint at a previously unreported sequestration insecticide resistance mechanism. If these protein families are indeed capable of binding and sequestering insecticides, it may be a characteristic of multiple members of each family as different transcripts were found elevated in assorted resistant populations. However, dsRNA silencing suggested only a single member of each family was involved in altering the resistance phenotype in the strain we examined. Although off-target siRNA effects cannot be excluded (and is examined in further detail in Supplementary Table 4), the large differences in

mortality seen within members of a single families indicates that this effect is not a major confounding factor in this study.

In addition to identifying putative effector genes linked to insecticide resistance (albeit by an as yet unknown mechanism) this study also implicated several transcription factors in regulating the pyrethroid resistance phenotype. Silencing of three separate transcription factors (*Dm*, *Met* and *Maf-S*) resulted in increased mortality after exposure to pyrethroid insecticides; using a data integration approach in this way can identify regulatory changes with small but consistent changes in fold change that were previously hard to detect by analysing individual data sets. The role of *Maf-S*, in regulating expression of metabolic resistance has been described previously[37] but it is noteworthy that approximately one third of the new resistance candidates identified in the current study also appear to be under the control of this transcription factor, elevating the status of this transcription factor to a key regulator of insecticide resistance in mosquitoes. Transcriptional silencing of *Met* also resulted in a significant reduction of expression in multiple detoxification family members, including *CYP6M2*, one of the key enzymes involved in insecticide metabolism[19,30], indicative of a role in pyrethroid resistance as well as the known role in juvenile hormone analogue resistance[46–48].

The production of the IR-TEx ShinyR app ensures that the insecticide resistance transcriptomic data are available and easily explored by all users without significant bioinformatics or programming experience. The value of this app in identifying potent novel insecticide resistance mechanisms, elucidating regulatory pathways and assigning putative functions and pathways to *Anopheles* transcripts has been demonstrated in this study. The simple nature of the data input sheet will allow users to add their own datasets, and for the datasets to be easily updated as

microarray experiments are published; hence increasing the power of the app. Further insights will accrue as the wider malaria vector community use this tool to interrogate their own datasets and validate additional candidates identified in this study.

## Methods

**Microarray datasets.** Literature and ArrayExpress searches were used to identify microarray experiments comparing resistant and susceptible populations of three Anopheline species: *An. gambiae*, *An. coluzzii* or *An. arabiensis*, across the African continent. Raw data files were acquired, from ArrayExpress[26] or VectorBase[27] (all data included before July 2017). All studies that were aimed at exploring the transcriptome of *An. gambiae* s.l. or *An. arabiensis* in relation to insecticide use were used in this study, no experiments were excluded. The resulting data were analysed using R[50] and all array designs were analysed as direct resistant vs susceptible experimental design. Within-array normalisation was carried out by loess, and between array normalisation by Aquantile[57]. Signals were corrected for dye by performing and correcting for dye swaps as directed in the limma package[23,24]. The limma package[23,24] was used to fit linear models to normalised corrected signals to assess differential expression, following the limma user guide on two-colour arrays with a common reference, using the functions lmFit and eBayes [https://bioconductor.org/packages/release/bioc/vignettes/limma/inst/doc/usersguide.pdf] for RvsS designs and a contrast matrix for complete loop designs. The limma outputs of each experiment were used to create a file used for the data integration by combining the fold changes and adjusted p values for each transcript across each experiment. Transcript fold changes and p values were averaged across each individual experimental data set if >75% of the probes: (a) showed the same fold change directionality or (b) were all non-significantly differentially expressed. All arrays used in this study compare two mosquito populations on an Agilent 8 × 15k *Anopheles gambiae* microarray platform (A-MEXP-2211). Transcripts with multiple probes (such as detoxification families) were averaged to produce a single value for fold change. For all transcripts of interest, BLAST of the probes to the *Anopheles gambiae* transcript database was used to identify potential cross-hybridisations. Subsetting of data-sets to answer data-driven hypotheses was performed using R, all significance levels are taken as adjusted p values of ≤0.05. A fold change of >1 was used to define up-regulation and <1 to define down-regulation.

**Variation calculation.** Metadata associated with each array was collected, including country, species, susceptible comparator, kdr levels, mortality levels, insecticide exposure and year collected. The variation explained by each factor was calculated using the unbiased estimator $\omega^2$.

$$\omega^2 = \frac{SS_{TREATMENT} - (\alpha - 1)MS_{S/A}}{S_{TOTAL} + MS_{S/A}}$$

SS = sum of squares, $\alpha$ = number of treatments, $MS_{S/A}$ = mean square error.

**Shiny app.** The fold change and Q values generated in data outputs from limma analysis were combined into a large dataset [see IR-Tex live: https://www.lstmed.ac.uk/projects/ir-tex or available for local use at github: https://github.com/LSTMScientificComputing/IR-TEx]. The resultant large dataset was used as the basis to develop a publicly accessible database. Using the package ShinyR[25], an online application was written to analyse transcripts of interest and output information on that transcript in each dataset, alongside strongly correlated transcripts both in tabular and graphical formats. Map display of significant transcripts was also integrated using the dismo package[58], The App 'IR-TEx' is available [https://www.lstmed.ac.uk/projects/ir-tex / https://github.com/LSTMScientificComputing/IR-TEx].

**Co-correlation.** Correlation networks were used to infer an association between transcripts across datasets using a pairwise correlation matrix as calculated in R using Pearson's correlation[50]. By manipulating high absolute Pearson correlation coefficient cut-off value (|r| > 0.75), stringency of the correlation network can be changed.

**Enrichment analysis.** Enrichment analysis was performed for all datasets on DAVID[52] and for detoxification transcripts/correlation network enrichments using a hypergeometric test on R (phyper), q = number of detoxification candidates present in a given list, m = number of detoxification candidates in the genome, n = number of transcripts in the genome – the number of detoxification family members and k = number of transcripts present in a given list. DroPhEA[49] was also used for enrichment analysis, using *Drosophila melanogaster* homologues. Benjamini-Hochberg multiple test correction was used throughout.

**Transcription factor search.** A list of putative and known Drosophila transcription factors were downloaded from flyTF.org[38] and *An. gambiae* orthologs identified using FlyMine[39]. The microarray database table [see IR-TEx Github URL:

https://github.com/LSTMScientificComputing/IR-TEx] was searched against these transcription factors and those differentially expressed in over 50% of the arrays (≥16) were extracted and visualised.

**Mosquito rearing conditions.** The *An. gambiae* s.l. used in these experiments were from the Tiassalé strain originally from Côte D'Ivoire, and the *An. coluzzii* were from VK7 and Banfora strains from Burkina Faso. All populations have been maintained under pyrethroid selection pressure in the insectaries at the Liverpool School of Tropical Medicine since 2009/2014/2014 respectively. The strains are resistant to pyrethroids and organochlorides[10,16,59,60]. The lab susceptible *An. coluzzii* population N'Gousso[61] was used as a qPCR comparitor. Mosquitoes were reared under standard insectary conditions at 27 °C and 70–80% humidity under a 12:12 h photoperiod and are presumed mated. Tiassalé was used for all RNAi experiments as it is the easiest of the resistant colonies to rear in large numbers; this colony was established in 2009 and originates from the same rice fields as the 'Tiassalé' microarray datasets (number of microarray datasets performed on Tiassalé = 4). VK7 and Banfora were colonised in 2014 and require arm-feeding and so are maintained at lower-levels; both populations have the same geographical origin as the microarrays with the resistant population same name (number of microarray datasets performed on VK7 = 2 and Banfora = 2).

**RT-qPCR.** RNA (4 μg) from each biological replicate was reverse transcribed using Oligo dT (Invitrogen) and Superscript III (Invitrogen) according to manufacturer's instructions. For the induction qPCR time course, alive adult female's RNA was extracted at 30 min, 1 h, 24 h and 48 h post-deltamethrin exposure. Quantitative real-time PCR was performed using SYBR Green Supermix III (Applied Biosystems) using an MX3005 and the associated MxPro software (Agilent). Primer Blast (NCBI)[62] was used to design primer pairs (Supplementary Table 8). Where possible, primers were designed to span an exon junction. Each 20 μl reaction contained 10 μl SYBR Green Supermix, 0.3 μM of each primer and 1 μl of 1:10 diluted cDNA. Standard curves were produced using whole N'Gousso cDNA, in 1, 1:5, 1:25, 1:125, 1:625 dilutions. qPCR was performed with the following conditions: 3 min at 95 °C, with 40 cycles of 10 s at 95 °C and 10 s at 60 °C. All amplification efficiencies of designed primers were within acceptable range (90–120%), following MIQE guidelines, relative expression was normalised against two housekeeping genes: EF and S7[63]. Validation of RNAi knockdown was performed after extraction of RNA from unexposed females 3 days post-injection. Analysis was performed on delta Ct values; Bartlett and Shapiro tests were used to confirm homogeneity of variance and normality of data respectively. For non-normal data square transformations were performed when possible. Normal data was analysed using an ANOVA followed by Dunnett's post hoc test, non-normal data was analysed using Kruskall–Wallis followed by a Dunn's post hoc test. Graphs were produced using GraphPad Prism 7. All qPCR analysis had three biological replicates and three technical replicates within each biological replicate.

**RNAi.** PCR was performed on Tiassalé cDNA using Phusion® High-Fidelity DNA Polymerase (Thermo Scientific) following manufacturer's instructions and primer sets with a T7 docking sequence at the 5′ end of both the sense and antisense primers. Primers were designed to produce an asymmetric product with a length of 300–600 bp, a GC content of 20–50% and no more than three consecutive equivalent nucleotides (Supplementary Table 9). PCR was performed with the following cycle: three minutes 98 °C, 35 cycles of seven seconds at 98 °C and 10 sec at 72 °C, with a final hold at 72 °C for seven minutes. PCR products were purified using a Qiagen QIAquick PCR Purification kit following manufacturer's instructions. dsRNA was synthesised using a Megascript® T7 Transcription (Ambion) kit, with a 16-hour 37 °C incubation, following manufacturer's instructions. The dsRNA was cleaned using a MegaClear® Transcription Clear Up (Ambion) kit, with DEPC water, twice heated at 65 °C for 10 min, to elute the sample. The resultant dsRNA product was analysed using a nanodrop spectrometer (Nanodrop Technologies, UK) and subsequently concentrated to 3 μg/μl using a vacuum centrifuge at 35 °C. 100, three-to-five day old, presumed mated, non-blood fed females, which were immobilised on a $CO_2$ block and 69 nl injected directly into the thorax, between the cuticle plates of the abdomen, underneath the wing. As a control, non-endogenous *GFP* dsRNA was injected at the same amount and concentration[64]. BLAST was carried out on all dsRNA constructs to identify off-site targets. Off-site targets are defined as sequences greater than 21 base pairs with 100% identity. dsRNAs constructs that resulted in greater than 40% reduction in mRNA abundance ($\mu = 78.8\% \pm 17.5\%$) were used for attenuation. Graphs were produced using GraphPad Prism 7.

**Bioassays.** 72-hours post injection, a minimum of 75 female mosquitoes were exposed to 0.05% deltamethrin impregnated papers using WHO bioassay tube test kits[65]. In each case, 25–30 treated females were present in each tube, the minimum number of replicates for each group is 3 (with the exception of *Dm*, where only two replicates were available). For each knockdown and each exposure, 20–25 female mosquitoes were simultaneously exposed to untreated papers as a control. Post-exposure, mosquitoes were left in a control tube, under insectary conditions for 24 h, with sucrose solution and mortality recorded. Analysis of mortality data was

done on normal data using an ANOVA test followed by a Tukey *post hoc* test. Graphs were produced using GraphPad Prism 7.

**Microarray experiment**. A whole-genome microarray approach was used to determine the effect of *Met* knockdown on transcriptional profiles. The transcriptional profiles of *Met* knockdowns were compared against a GFP injected control. RNA was extracted from three biological replicates for each of *Met* injected and GFP injected controls. Mosquitoes were collected 72 h post injection, between the hours of 8am and 2 pm. In both cases larvae were collected and reared in insectary conditions and species ID[66] performed on the colony. Each replicate was added to extraction buffer from the PicoPure RNA extraction kit, heated for 30 min at 42 °C and frozen at −80 °C as per manufacturer's instructions. Each biological replicate for each treatment consisted of RNA, extracted using PicoPure RNA Isolation kit (Arcturus), from 7–12 three to five-day-old non-blood fed, presumed mated females. The quantity and quality of the RNA was assessed using a nanodrop spectrophotometer (Nanodrop Technologies UK) and Bioanalyser (Agilent) respectively. 100 ng of RNA was amplified and labelled with Cy3 and Cy5, using the Two colour low input Quick Amp labelling kit (Agilent) following the manufacturer's instructions. Samples were then purified (Qiagen) with the cRNA yield and quality assessed using the nano-drop and Bioanalyser respectively. RNA from each *Met* injection replicate was competitively hybridised with the GFP injected control replicates. Dye swaps were performed on each of the technical replicates for each array, to correct for dye bias. Labelled cRNAs were hybridised to the whole genome 8 × 15k *Anopheles gambiae* array (ArrayExpress accession number A-MEXP-2211). Microarray hybridisation, washing and scanning were performed according to previously described protocols[30]. The *dsMet* experiment was submitted to ArrayExpress, accession E-MTAB-4043. Banfora, Tiefora 2013 and Tiefora 2014 microarrays were submitted to ArrayExpress[26] accession numbers: E-MTAB-6498, E-MTAB-6499, E-MTAB-6500 respectively, with all relevant information on study design. All data were analysed using the limma package in R, with background and between- and within- array normalisations as previously reported[53].

**Code availability**. All code used in this study is available on the IR-TEx Github (https://github.com/LSTMScientificComputing/IR-TEx).

## Data availability
The datasets generated during the current study are available on ArrayExpress under the accession numbers: E-MTAB-4043, E-MTAB-6498, E-MTAB-6499 and E-MTAB-6500. All data analysed during this current study are available on public repositories and detailed in the present paper, as detailed in Supplementary Table 1. The authors declare that all other data supporting the findings of this study, are available within the article and its Supplementary Information files, or are available from the authors upon request.

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

## Acknowledgements

This study was funded by an MRC Skills Development Fellowship (to V.I.) and a Royal Society Challenge Grant (to H.R.). We are grateful to Dr Jonathan Moore for advice on data analysis, to Manuela Bernardi for help with preparation of the figures and to Andrew Bennett and Dr James Maas for work on hosting IR-TEx.

## Author contributions

V.I. and H.R. designed the experiments and the IR-TEx app, V.I. completed all experiments and coding for IR-TEx and analysed the data. V.I. and H.R. drafted the manuscript, S.W. provided support for the development of the IR-Ex app.

## Additional information

**Competing interests:** The authors declare no competing interests.



