## [Peer Review File · Nature Communications]

Reviewers' Comments:

Reviewer #1:

Remarks to the Author:

This study describes a meta-analysis of existing gene expression data to identify and explore genes potentially involved in resistance to pyrethroids in mosquito vectors of malaria. This is combined with additional functional and expression analysis. The manuscript provides significant new insights into the molecular basis of resistant mosquito populations in Africa uncovering previously unknown mechanisms of resistance. This novelty is combined with a very practical output in the form of a user-friendly app that can be used to interrogate candidate resistance genes or new data-sets. In combination this study represents an extremely valuable and novel contribution to the literature and is of significant practical utility – I have little doubt that it will become a highly cited paper.

A few minor comments below:

Figure 1 is not as easy to follow as it could be – the arrow/circle system to indicate phenotype is confusing and unintuitive – I suggest a traffic light system is used to phenotype (use a grey shade for unknown) with species indicated in some other way (initials Aa, Ag etc.?).

Figure 2 –it would be informative to see the susceptible strain included in all qPCR bar charts to gauge variation in expression between this strain and the three resistant populations. In this context standard error bars are uninformative and should be replaced with standard deviation or even better 95% CLs. Finally I don't think a t-test is appropriate for this analysis as you are comparing multiple populations – an ANOVA followed by post-hoc testing is more appropriate (performed on delta CTs – not delta/delta CTs).

Figure 3 – see my comment on standard error above.

There is a sentence fragment in line 477: Injections were carried out on.

Reviewer #2:

Remarks to the Author:

In this submission, the authors aimed to identify novel genes or pathways involved in insecticide resistance of malaria vectors via performing a meta-analysis on Anopheles transcriptomes. This meta-analysis was conducted through the integration of 31 microarray datasets that profiled mRNA expression of 22 geographically isolated mosquito populations. Based on the results presented, the authors claimed that they have identified several previously unreported candidate genes underlying insecticide resistance. The possible mechanisms of some of these genes (e.g. alpha-crystallins, hexaerins, Met) were discussed. In addition, the authors have developed a web application, IR-TE_x, for the users to investigate the data compiled and processed. Although this study is indeed the first study trying to integrate and analyze transcriptomes of sub-Saharan insecticide resistant mosquitos simultaneously, the approach used is conventional and the presentation of the results are largely descriptive. There are also some issues that could potentially affect the interpretation of the results unaddressed and undiscussed, including issues related with using microarray to approximate expression signals of genes with paralogs in the genome, and issues related with specificity of miRNAs that are used to validate the results. As for the web application IR-TE_x, I found that at least the instructions for the users and the readability of outputs should be improved. I also feel that the clarity of the manuscript should be improved. Because of the abovementioned issues, I am hesitant to recommend acceptance of this submission for publication in the present format. Below please see my

specific comments in detail.

1. "Issues of using microarray-based mRNA expression data to identify the causal genes underlying the phenotype of interest": The accuracy of microarray signals is determined by probe-target specificity. The design of probes for genes with a closely related paralogs in the genome or genes with a large number of paralogs in the genome is particularly challenging because these probes tend to be subject to non-specific binding from mRNAs transcribed from paralogs. The authors reported seven genes in GST, CYP and COE families are likely candidate genes for pesticide resistance but were previously overlooked; however, as noted by the authors, some genes in these three families have been known to be capable to detoxify insecticides (refs 7-18), and the observed hybridization signals of the 7 candidate genes that are partially contributed by the known genes were not controlled or removed. The authors should at least evaluate if the probes for the candidate genes identified in the present study are subject to cross-hybridization, and this analysis could potentially explain why the case of AGAP013061 (the second panel of Figure 2) could not be validated by qPCR.

2. "Mysterious patterns potentially related with RNAi specificity": I had difficulty in understanding the logical flow of the section alpha-crystallins (lines 159-185). Although increased expression of AGAP007161 was observed in multiple populations (Figure 2), RNAi knockdown of this gene did not cause increased mortality in comparison with the control (Figure 3a). This actually suggested that AGAP007161, the one that showed the most consistent patterns identified by the approach, is a false positive. Which data prompted the authors to look at clustered gene family instead of the single gene while investigating alpha-crystallins? It is unclear how another alpha-crystallin gene AGAP007159 that is more associated with the resistant phenotype (Figure 3) was not identified and included in Tables S2-S3 and Figure 2? Could this mysterious observation be related with RNAi specificity, which was not evaluated in this study?

3. "The web application IR-TEEx needs to be substantially improved": First of all, a clear instruction or tutorial should be provided to the users. It is unclear in IR-TEEx which ID system for the data input is supported. Although it is claimed the users can feed IR-TEEx with their own data sets to compare with other mosquito populations, it was not demonstrated how such a task could be performed. Multiple examples with clear explanations and demonstrations are required to be provided on the web. Second, the output used abbreviations in describing the populations, making it very inconvenient for the users to understand the results. Because the input gene could show elevated expression in some populations but suppressed expression in the others, I suggest that some statistics in testing if the selected gene shows significant expression change based on the selected populations as a whole could be performed. Third, it takes a while for IR-TEEx to produce the result even when only one gene is given for the analysis (I do not know how to analyze multiple genes in IR-TEEx). The speed should be improved, maybe achieved by conducting the analysis based on intermediate files precomputed from the primary datasets.

4. "The clarity of this manuscript should be enhanced": Below are some suggested places to be improved: (i) To describe genes, the authors sometimes use gene symbols, while sometimes used full gene names or Ensembl IDs. I suggest that gene symbols should be provided in Figure 2, and Ensembl IDs should be provided along genes symbols mentioned between lines 125 and 126. (ii) Accession numbers of the data source (e.g. E-MTAB-xxxx, ArrayExpress) should be described in Table S1. (iii) Statistics should be performed for Figures 3c and 4b. (iv) Statistical values for enrichment on GO terms or phenotypes should be provided. It is unclear if the P-values described have been controlled for multiple testing. If yes, which method was used. (v) Most of the Supplementary Tables are not readable and unorganized. If the tables are large, they should be provided as EXCEL files (such as the file "168256_0_supp_3077851_p96scm") instead of being provided as PDF files.

Reviewer #3:

Remarks to the Author:

Ingham and colleagues present a meta analysis of microarray data on insecticide resistance in 3 major vectors of malaria in sub-Saharan Africa. The authors confirm previously known resistance candidates and also identify novel candidates, some of which they do further work on via RNAi mediated gene silencing experiments. They also look at transcriptional regulators and do validation by gene silencing. Finally they have developed an interactive Web based application that allows one to explore genes of their own interest, identifies suites of genes with shared expression patterns and can be used to further explore novel candidates for insecticide resistance.

Certainly, the area of insecticide resistance is of great importance to the mosquito research community and going beyond the canonically studied insecticide resistance genes is key to our understanding of mechanisms at work in wild populations. Ingham and colleagues chose the meta-analysis approach and I have some concerns about their approach largely driven by the dearth of detail provided.

Overall, given that meta-analysis is a central tenet of this manuscript, I find the detail behind the analytical approach they used to be lacking. As described by Bioconductor and in the Smyth paper (#23 cited by the authors), limma is not a Bioconductor/R package designed to do meta-analysis of microarray data, but instead, it is my understanding that it uses a linear model approach to assess differences in gene expression without specific tools for handling the combination of data across disparate studies. There is an updated publication on limma, Ritchie et al, limma powers differential expression analyses for RNA-sequencing and microarray studies. Nucleic Acids Research 2015 which may be a more appropriate citation or an additional citation, but also does not focus on meta-analysis of microarray data.

If the authors identify limma as the right tool for meta-analysis, there needs to be adequate justification, particularly given other options exist and much greater clarity about how the analysis was done. Specially, the authors need to address the myriad of issues that arise when doing meta-analysis, including, but not limited to, differences in platform (Affy, Illumina), differences in study design and study aim, inherent differences across labs. Though certainly not an exhaustive list, CrossMeta, GeneMeta and RankProd are all Bioconductor packages whose aim is meta-analysis. If limma is appropriate for the meta-analysis performed here, the authors must justify this and give much greater detail on their analytical approach in the methods. Further, given the importance of meta-analysis to this paper, I think parameters need to be greater explored and explained, not described as default as on line 405.

In general, there are acknowledged issues with meta-analysis, regardless of data type, including differences in study aims and study design. When microarray data are involved, there are additional issues to consider including challenges with different probes and probe sets, different platforms and lab to lab variation. Along these lines, and not addressed by the authors, was the meta-analysis done at the gene level or the probe level. If a single study lacked data for a given probe or gene, how was this handled? And how might this bias the results?

A very general question pertaining to the meta-analysis, does it simply provide greater statistical power given the greater sample size? Or in other words, what advantage does the meta-analytical approach have over taking gene lists from each of the individual papers and comparing these?

How were the 31 studies chosen, strictly on a literature search? Do other studies exist that were excluded? Or were all studies that meet a certain set of criteria (that should be spelled out in greater detail) used? What types of arrays were used in these studies (whole genome vs candidate gene, Affy, Illumina, Nimblegen) (this information may be a useful addition to Supplementary Table 1).

Given the Tiassale and VK7 colonies are different mosquito species (Tiassale is *A. gambiae* and VK7 is *A. coluzzii*, if I recall correctly), why in Figure 3 for instance are only those genes that gave significant results in *A. gambiae* tested in *A. coluzzii*? Are the authors only interested in insecticide resistance that is shared across species? I would argue that it is important to understand both species general and species specific insecticide resistance as both of those pieces of information would greatly arm the vector control tool box. Along these lines, provided greater background on these colonies and their history would be helpful and also an explanation of why some colonies are used for all experiments and some only for one experiment would be helpful.

With regard to mosquito species, Figure 1 makes it pretty obvious, that in the studies used, there is a confounding of species and geography and a large difference in resistance across the species. How are these factors dealt with in your study?

All of the figures for gene silencing experiments should include sample size (n) information on the plot?

With the increase in RNASeq and the decline in microarray studies, would the interactive web application be able to accept RNASeq data and have a method for handling and comparing it to microarray data?

Other comments

To help with organization and presentation, I would present lines 97-108 in one section and lines 110-117 in a separate section with its own heading.

Line 116-117—the authors state that there is little overlap between the two lists, this is easily quantified and should be presented.

Lines 132-133, are CYP6M2 and CYP6P3 reliably measured in all of the 11 studies across different microarray platforms?

Lines 140-142, why have these genes been previously overlooked, is it due to the candidate gene focus of many studies or due to a lack of statistical power within individual studies?

Line 146—the species of the 3 colonies should be specified. Tiassale is *A. gambiae*, I think and the other 2 are *A. coluzzii*. Also it would be worth mentioning whether the origin of any of these colonies are sympatric with any of the gene expression studies.

It would be informative to see how the genes that appear in Supplementary Tables 2 and 3 were measured in individual studies, what fold changes and p values did they have in individual studies as compared to the meta-analysis.

Line 87 says 22 geographically distinct mosquito populations, but Figure 1 only has 21 circles.

Further it would be nice to highlight in Figure 1 where the 12 datasets used in the first analysis are from. Along these same lines—when presenting a new analysis, it would be helpful to clearly spell out

for the reader which of the 31 studies are included and how inclusion of particularly subsets of studies provides the best statistical power to address the hypothesis posed.

In figures like Figure 2, it would be helpful to note in the legend that not all the y axes are the same.

Figure legends for RT-PCR data must include numbers of replicates, n or something to understand where the error bars come from.

For the heat maps—what is the order of the locations on the x axis? Geographic? Or put another way, why are they shown in the order that they are?

Y axes need to be labelled, not just mentioned in the figure legend.

Editor Comments:

..... a revised manuscript would need to address the concerns on confounding factors in the data, the microarray-based meta-analysis, siRNA off-target effects, the web application, and statistics. Please make sure that these and all other raised concerns are addressed in full in a revised manuscript.

As requested by the editor, the manuscript has been substantially reviewed in all areas:

1. A 9-page user guide, dealing with many of the reviewer's questions has been provided as a supplementary file. Within this guide there are instructions on: (i) How to use the app and what the outputs of the app represent; (ii) How to input other insecticide resistance datasets performed on any kind of -omics platform and what to do in the case of differing probes from different arrays; (iii) How to edit the code to use the app with unrelated datasets (any organism, any field) and (iv) How to integrate different -omics platforms with the app with an example dataset that comes from use of this app as part of the Catteruccia Group at Harvard T.H Chan School of Public Health.
2. The code of the app has been optimised, after comments on the slow load speed. The app previously loaded in around 10 seconds on LSTM's server, the new code will load in 2/3 of the time; this will improve usability of the app and make it more appealing to users. The calculation of a 15k x 15k matrix is the confounding step; one calculation of this matrix has been removed (previously, there were three iterations, now there are two).
3. All statistics have been re-done and the figures edited appropriately.
4. Both RNAi and Microarray are subject to confounding by cross hybridisation. In order to address the risk of this occurring within this study, each microarray probe of transcripts of interest was searched against the Anopheles transcript database using BLAST to test for any cross-hybridisation. Where the potential for cross hybridisation occurs, it has been reported both in a new Supplementary Table 4 and also within the text, with a discussion each time on how this might impact the interpretation of the results. Similarly, each siRNA construct was informatically split into 21bp segments which were used to BLAST against the Anopheles transcript database, where there was 100% identity it is reported in Supplementary Table 4. Again, any impact on the overall story of the paper has been discussed in the text.
5. The methods have been re-written to improve clarity, and hopefully with the inclusion of a quantitative statement on the variability of the dataset, we have addressed reviewer 3's question regarding use of a standard meta-analysis package.
6. Finally, we realise that the term 'meta-analysis' can have different meanings in different disciplines. Our definition of the phrase meta-analysis may not align with that of reviewer 3 and so, dependent upon the editor's opinion, we could remove this term in the title and throughout the paper.

Reviewer 1:

Figure 1 is not as easy to follow as it could be – the arrow/circle system to indicate phenotype is confusing and unintuitive – I suggest a traffic light system is used to phenotype (use a grey shade for unknown) with species indicated in some other way (initials Aa, Ag etc.?).

Figure 1 and legend have been replaced with the suggested edits.

Figure 2 –it would be informative to see the susceptible strain included in all qPCR bar charts to gauge variation in expression between this strain and the three resistant populations. In this context standard error bars are uninformative and should be replaced with standard deviation or even better 95% CLs. Finally I don't think a t-test is appropriate for this analysis as you are comparing multiple populations – an ANOVA followed by post-hoc testing is more appropriate (performed on delta CTs – not delta/delta CTs).

Figure 3 – see my comment on standard error above.

All qPCR graphs have been changed to show the comparator, such as susceptible or unexposed mosquitoes to allow variance to be seen. Standard error bars have been replaced with standard deviation in each case. The statistics have been re-done on delta Ct values: an ANOVA followed by a Dunnett's *post hoc* test for normal data (as tested by Shapiro/Bartlett tests), if transformation cannot achieve normality and Kruskal-Wallis followed by a Dunn's *post hoc* test have been used.

There is a sentence fragment in line 477: Injections were carried out on.

Corrected

Reviewer 2:

1. "Issues of using microarray-based mRNA expression data to identify the causal genes underlying the phenotype of interest": The accuracy of microarray signals is determined by probe-target specificity. The design of probes for genes with a closely related paralogs in the genome or genes with a large number of paralogs in the genome is particularly challenging because these probes tend to be subject to non-specific binding from mRNAs transcribed from paralogs. The authors reported seven genes in GST, CYP and COE families are likely candidate genes for pesticide resistance but were previously overlooked; however, as noted by the authors, some genes in these three families have been known to be capable to detoxify insecticides (refs 7-18), and the observed hybridization signals of the 7 candidate genes that are partially contributed by the known genes were not controlled or removed. The authors should at least evaluate if the probes for the candidate genes identified in the present study are subject to cross-hybridization, and this analysis could potentially explain why the case of AGAP013061 (the second panel of Figure 2) could not be validated by qPCR.

A supplementary table (Supplementary Table 4) has been added to illustrate potential cross hybridisation between the candidate genes described in the text and any other transcripts in the *An. gambiae* genome. The vast majority of the probes are specific to their target gene. The potential for co-hybridisation between two p450s (CYP6Z2 and CYP6Z3) has been mentioned in the figure legend of Supplementary Figure 1, as well as the potential for CYP12F2 to cross hybridise to an unnamed p450 (lines 135-136). The non-validation of some candidates via qPCR is likely due to the mosquitoes used being unexposed mosquitoes in order to identify ubiquitous over expression, whereas the majority of the microarray datasets are exposed to insecticide; this is now mentioned in the text (lines 158-160).

2. "Mysterious patterns potentially related with RNAi specificity": I had difficulty in understanding the logical flow of the section alpha-crystallins (lines 159-185). Although increased expression of AGAP007161 was observed in multiple populations (Figure 2), RNAi knockdown of this gene did not cause increased mortality in comparison with the control (Figure 3a). This actually suggested that AGAP007161, the one that showed the most consistent patterns identified by the approach, is a false

positive. Which data prompted the authors to look at clustered gene family instead of the single gene while investigating alpha-crystallins? It is unclear how another alpha-crystallin gene AGAP007159 that is more associated with the resistant phenotype (Figure 3) was not identified and included in Tables S2-S3 and Figure 2? Could this mysterious observation be related with RNAi specificity, which was not evaluated in this study?

AGAP007161 could be a false positive or could be a non-crucial component in an insecticide resistance pathway or have redundancy with other a-crystallins; this is now discussed in the text (lines 202-203). We decided to investigate all 8 members of the alpha-crystallin family due to the tight clustering and up-regulation of several of these in the microarray experiments, with the hypothesis that they may be transcriptionally co-regulated; this is now clarified in the text (lines 177-180). AGAP007159 and AGAP007158 probes are identical and so there will be cross-hybridisation, this has now been noted in the text (lines 190-193) and in the new Supplementary Table 4. RNAi specificity has now been addressed in the new supplementary table 4 and is mentioned in the discussion (lines 394-397) – although we realise that siRNA off-target effects are an inevitable limitation of this technology, the clear differences in phenotype observed between members of a gene family support our interpretation that this is not an important confounding factor in this analysis.

3. “The web application IR-TEx needs to be substantially improved”:

First of all, a clear instruction or tutorial should be provided to the users. It is unclear in IR-TEx which ID system for the data input is supported. Although it is claimed the users can feed IR-TEx with their own data sets to compare with other mosquito populations, it was not demonstrated how such a task could be performed. Multiple examples with clear explanations and demonstrations are required to be provided on the web.

A new in depth-user guide has been written and has been submitted alongside the manuscript (Supplementary File 1); it will also be available from the github repository. The user guide now contains walk throughs on adding new resistance data, modifying the app for use with non-resistance data and information on how to integrate other transcriptomic data such as RNAseq and one-colour arrays. It is also made clear in this guide that transcript ID is to be used within the app.

Second, the output used abbreviations in describing the populations, making it very inconvenient for the users to understand the results.

This is unfortunately necessary for data display; the populations are also arbitrarily named following the published manuscripts. Supplementary Table 1 is provided to explain full meta-data on all the populations included.

Because the input gene could show elevated expression in some populations but suppressed expression in the others, I suggest that some statistics in testing if the selected gene shows significant expression change based on the selected populations as a whole could be performed.

We apologise if we are missing the point of the reviewer’s suggestion, but this tool is intended to explore transcripts of interest with the necessity for *post hoc* experimental validation. In addition to this, there are intrinsic difference within species, such as exposure status, geographical location and susceptible comparator that would make this statistic hard to interpret.

Third, it takes a while for IR-TE_x to produce the result even when only one gene is given for the analysis (I do not know how to analyze multiple genes in IR-TE_x). The speed should be improved, maybe achieved by conducting the analysis based on intermediate files precomputed from the primarily datasets.

Due to the large size and the reactive nature of calculating correlation networks, it is not possible to load these tables directly and hence save time on loading. We have reduced the load time from around 10 seconds by around 3.5 seconds by code optimisation of the correlation calculation. VectorBase are also interested in hosting the app and with their development team, the speed should be improved. It is also possible to improve speed drastically by running the app locally, which is described in the user guide (<4s on a standard i7 intel machine).

4. "The clarity of this manuscript should be enhanced": Below are some suggested places to be improved:

(i) To describe genes, the authors sometimes use gene symbols, while sometimes used full gene names or Ensembl IDs. I suggest that gene symbols should be provided in Figure 2, and Ensembl IDs should be provided along genes symbols mentioned between lines 125 and 126.

As many of these transcripts don't have unique gene names, gene IDs have been provided throughout for consistency.

(ii) Accession numbers of the data source (e.g. E-MTAB-xxxx, ArrayExpress) should be described in Table S1.

Accession numbers for all datasets are present in Supplementary Table 1

(iii) Statistics should be performed for Figures 3c and 4b.

Now addressed in the text.

(iv) Statistical values for enrichment on GO terms or phenotypes should be provided. It is unclear if the P-values described have been controlled for multiple testing. If yes, which method was used.

These are now provided throughout, multiple test correction method added to methods (lines 479-480).

(v) Most of the Supplementary Tables are not readable and unorganized. If the tables are large, they should be provided as EXCEL files (such as the file "168256_0_supp_3077851_p96scm") instead of being provided as PDF files.

All supplementary tables are Excel files but they may have been converted to pdfs by the journal for the review process

Reviewer 3:

Overall, given that meta-analysis is a central tenet of this manuscript, I find the detail behind the analytical approach they used to be lacking. As described by Bioconductor and in the Smyth paper (#23 cited by the authors), limma is not a Bioconductor/R package designed to do meta-analysis of microarray data, but instead, it is my understanding that it uses a linear model approach to assess differences in gene expression without specific tools for handling the combination of data across disparate studies. There is an updated publication on limma, Ritchie et al, limma powers differential

expression analyses for RNA-sequencing and microarray studies. Nucleic Acids Research 2015 which may be a more appropriate citation or an additional citation, but also does not focus on meta-analysis of microarray data.

The reviewer is correct in the assumption that limma was not used for the meta-analysis itself but for the analysis of each individual experiment to give fold change values between resistant and susceptible mosquitoes. To address these concerns, the methods have been clarified and the additional citation has been added to the work (lines 425-447).

If the authors identify limma as the right tool for meta-analysis, there needs to be adequate justification, particularly given other options exist and much greater clarity about how the analysis was done. Specially, the authors need to address the myriad of issues that arise when doing meta-analysis, including, but not limited to, differences in platform (Affy, Illumina), differences in study design and study aim, inherent differences across labs. Though certainly not an exhaustive list, CrossMeta, GeneMeta and RankProd are all Bioconductor packages whose aim is meta-analysis. If limma is appropriate for the meta-analysis performed here, the authors must justify this and give much greater detail on their analytical approach in the methods. Further, given the importance of meta-analysis to this paper, I think parameters need to be greater explored and explained, not described as default as on line 405.

The reviewer is mistaken here. As explained above, we did not use limma for the meta-analysis. We apologise if this was not made clear in the original submission. The methods have been clarified. The application user guide has also been updated and now contains steps on how to account for differences in -omic platform through applying quantile normalisation to the gene expression outputs. In the present study, the datasets are all performed on the same platform, with similar resistant vs susceptible design; this is now clarified in the methods. The methods have been changed to add detail to the parameters used in each instance (lines 425-447). The use of a data integration approach rather than meta-analysis packages was taken due to the small amount of variation explained by known meta-data and hence 'noisiness' of the dataset – this has been clarified in the results section (lines 93-95).

In general, there are acknowledged issues with meta-analysis, regardless of data type, including differences in study aims and study design. When microarray data are involved, there are additional issues to consider including challenges with different probes and probe sets, different platforms and lab to lab variation. Along these lines, and not addressed by the authors, was the meta-analysis done at the gene level or the probe level. If a single study lacked data for a given probe or gene, how was this handled? And how might this bias the results?

The study was done on the transcript level. For genes in which there were multiple probes on the array, signals from across the probes was averaged; this is now clarified in the methods section (lines 425-447). The other questions are now answered in the user guide in the section addressing using different platforms and different -omics approaches.

A very general question pertaining to the meta-analysis, does it simply provide greater statistical power given the greater sample size? Or in other words, what advantage does the meta-analytical approach have over taking gene lists from each of the individual papers and comparing these?

Again, we apologise if we have misunderstood this comment but this is exactly what we did. We did compare gene lists, rather than run a meta-analysis program. We did this due to the high amount of 'noise' in the data-sets; to minimise this, we analysed all the microarray datasets using

the same methodology (limma) and then applied hypothesis driven approaches such as looking at only pyrethroid exposed populations or looking at expression of specific gene families such as P450s and transcription factors. As mentioned to the editors, we believe our use of the word ‘meta-analysis’ may have different connotations dependent upon the reader and could change this in the title and throughout the text.

How were the 31 studies chosen, strictly on a literature search? Do other studies exist that were excluded? Or were all studies that meet a certain set of criteria (that should be spelled out in greater detail) used? What types of arrays were used in these studies (whole genome vs candidate gene, Affy, Illumina, Nimblegen) (this information may be a useful addition to Supplementary Table 1).

All arrays used in this study were done on the same Agilent platform; this was not a criterion of the literature search, rather the use of a single Agilent array, that has multiple probe for detoxification family genes, is ubiquitous in insecticide resistance studies on these mosquito species. This has been specified in the methods section (lines 440-441).

Given the Tiassale and VK7 colonies are different mosquito species (Tiassale is *A. gambiae* and VK7 is *A. coluzzii*, if I recall correctly), why in Figure 3 for instance are only those genes that gave significant results in *A. gambiae* tested in *A. coluzzii*? Are the authors only interested in insecticide resistance that is shared across species? I would argue that it is important to understand both species general and species specific insecticide resistance as both of those pieces of information would greatly arm the vector control tool box. Along these lines, provided greater background on these colonies and their history would be helpful and also an explanation of why some colonies are used for all experiments and some only for one experiment would be helpful.

The selection of the Tiassale colony for the first round of dsRNAi experiments was a practical choice. The Tiassale is actually a mixture of ‘M’ and ‘S’ form mosquitoes ie. *An. gambiae* and *An. coluzzii*. Tiassale is used in many experiments as it is the most robust of our resistant populations and simple to rear in large numbers in the insectary. VK7 was colonised more recently and it is more challenging to rear in very large numbers. Hence we tested the full set on Tiassale and then only those that were showed a significant phenotype were tested on VK7. These details are now included in the methods section (lines 496-502). We agree it is extremely important to understand both the similarities and differences in the resistance mechanisms of these species and we hope that, by making a publically available, user friendly app, we will help address this question as more microarray data sets become available.

With regard to mosquito species, Figure 1 makes it pretty obvious, that in the studies used, there is a confounding of species and geography and a large difference in resistance across the species. How are these factors dealt with in your study?

Variation due to geographical difference/resistance levels accounted for a tiny amount of the variation in the dataset; combined all known meta-data accounted for less than 0.5% of the variation; this is now discussed (lines 93-95). To avoid these problems, we used hypothesis driven approaches, such as looking at only pyrethroid exposed populations, as mentioned above.

All of the figures for gene silencing experiments should include sample size (n) information on the plot?

Added

With the increase in RNASeq and the decline in microarray studies, would the interactive web application be able to accept RNASeq data and have a method for handling and comparing it to microarray data?

Yes, and information on how to do this is included in the new user guide.

Other comments

To help with organization and presentation, I would present lines 97-108 in one section and lines 110-117 in a separate section with its own heading.

Done

Line 116-117—the authors state that there is little overlap between the two lists, this is easily quantified and should be presented.

Added a sentence to show the minimal overlap – just 2 up-regulated transcripts present in each list (lines 124-125).

Lines 132-133, are CYP6M2 and CYP6P3 reliably measured in all of the 11 studies across different microarray platforms?

Yes; this has been discussed a little more in the methods regarding platform and also in probe alignment as per Reviewer 2's comments (Supplementary Table 4).

Lines 140-142, why have these genes been previously overlooked, is it due to the candidate gene focus of many studies or due to a lack of statistical power within individual studies?

It is due to *a priori* candidates often being picked out of the list; this is now mentioned.

*Line 146—the species of the 3 colonies should be specified. Tiassale is *A. gambiae*, I think and the other 2 are *A. coluzzii*. Also it would be worth mentioning whether the origin of any of these colonies are sympatric with any of the gene expression studies.*

This is clarified in the methods (see also response to earlier question).

It would be informative to see how the genes that appear in Supplementary Tables 2 and 3 were measured in individual studies, what fold changes and p values did they have in individual studies as compared to the meta-analysis.

Fold changes have been added to the supplementary tables; we have not included the p values for each gene in each population but this can be readily searched using the app.

Line 87 says 22 geographically distinct mosquito populations, but Figure 1 only has 21 circles.

VK7 and VK6 are neighbouring villages but still geographically distinct – Figure 1 has been changed to include both for clarity.

Further it would be nice to highlight in Figure 1 where the 12 datasets used in the first analysis are from. Along these same lines—when presenting a new analysis, it would be helpful to clearly spell out for the reader which of the 31 studies are included and how inclusion of particularly subsets of studies provides the best statistical power to address the hypothesis posed.

These data sets have been highlighted on the new Figure 1, with dotted lines.

In figures like Figure 2, it would be helpful to note in the legend that not all the y axes are the same.

The figures have been edited so that the y axes all have the same range on the ‘lower’ axes (split axes are used where needed)

Figure legends for RT-PCR data much include numbers of replicates, n or something to understand where the error bars come from.

Included.

For the heat maps—what is the order of the locations on the x axis? Geographic? Or put another way, why are they shown in the order that they are?

Y axes need to be labelled, not just mentioned in the figure legend.

The arrays are in order that they appear in the IR-TEx app and Supplementary Table 1; this changes in the app dependent upon the selections of the user. The y axes have now been labelled.

Reviewers' Comments:

Reviewer #1:

Remarks to the Author:

The authors have satisfactorily addressed all my comments - thus I recommend it is now accepted.

Reviewer #2:

Remarks to the Author:

The authors have made substantial efforts in addressing my comments. I have no further suggestion except for the following two minor points for the IR-TE_x app developed in this study: (a) The color used to represent the range of values selected in "Absolute Correlation Value" is counterintuitive. I suggest that the selected region should be colored, and the unselected regions should be uncolored. (b) The font used in the X-axis of dashboards "expression line graph" and "correlation line graph" could be enlarged and aligned vertically to improve the readability.

Reviewer #3:

Remarks to the Author:

In general the authors have addressed the concerns I raised during my initial review as well as concerns raised by other reviewers and I feel that the manuscript is markedly improved. I would agree with the authors that the term meta-analysis may not have a coherent definition however, for that reason, it either needs to be defined clearly by the authors to avoid confusion and assumptions or, as they suggest, it needs to be removed.

To me, meta-analysis is a statistical analysis that combines the results of multiple independent studies. One of the basic ideas behind meta-analysis is that there is a common truth behind similar scientific studies which has been measured with some error in each individual study. The aim of the meta-analysis approach is use approaches from statistics to derive a pooled estimate closest to the real truth based on these errors. Many meta-analysis methods produce a weighted average computed from the results of individual studies taking into account error from individual studies.

Related to the meta-analysis done here, the authors state that they combined fold changes and adjusted p value for each transcript across each experiment. How was this done, with weighted averages? Even if the word meta-analysis is removed, greater detail is required here on how these values were combined.

Lastly, in Figure 3 the legend states that 3 biological and 3 technical replications were done for RNAi experiments and Bioassays. What does this mean and how were these handled for the statistics— some replication at the level of RNAi injection and some at the level of Bioassays? Is this 3 technical replications within each biological replicate? Presumably the sample sizes are combined across all these replicates? Very little experimental detail on replicates is given in the methods.

Editors Comments:

In particular, please address the concerns of reviewer 2 on the IR-TEx app, provide the information requested by reviewer 3, and avoid the term 'meta-analysis' throughout the manuscript. Please highlight all changes in the manuscript text file.

As requested by the editor, these issues have now been addressed and are expanded on in the reply to the reviewers below.

Reviewer #1 (Remarks to the Author):

The authors have satisfactorily addressed all my comments - thus I recommend it is now accepted.

Reviewer #2:

The authors have made substantial efforts in addressing my comments. I have no further suggestion except for the following two minor points for the IR-TEx app developed in this study: (a) The color used to represent the range of values selected in "Absolute Correlation Value" is counterintuitive. I suggest that the selected region should be colored, and the unselected regions should be uncolored. (b) The font used in the X-axis of dashboards "expression line graph" and "correlation line graph" could be enlarged and aligned vertically to improve the readability.

We have made changes to the app as suggested by reviewer two. There is now a slider, so the selected area is coloured in blue. We have also changed the alignment on the line graphs within the app as suggested.

Reviewer #3:

In general the authors have addressed the concerns I raised during my initial review as well as concerns raised by other reviewers and I feel that the manuscript is markedly improved. I would agree with the authors that the term meta-analysis may not have a coherent definition however, for that reason, it either needs to be defined clearly by the authors to avoid confusion and assumptions or, as they suggest, it needs to be removed.

Meta-analysis has been removed throughout the text.

Related to the meta-analysis done here, the authors state that they combined fold changes and adjusted p value for each transcript across each experiment. How was this done, with weighted averages? Even if the word meta-analysis is removed, greater detail is required here on how these values were combined.

We have clarified further the methodology behind the combining of the fold changes on lines 438-440.

Lastly, in Figure 3 the legend states that 3 biological and 3 technical replications were done for RNAi experiments and Bioassays. What does this mean and how were these handled for the statistics—some replication at the level of RNAi injection and some at the level of Bioassays? Is this 3 technical replications within each biological replicate? Presumably the sample sizes are

combined across all these replicates? Very little experimental detail on replicates is given in the methods.

In each case in which replicates are used, the methodology has been clarified in the text (lines 214-215, lines 274-275, lines 521-522, lines 546-547)